# Potential Use of Edible Insects in Complementary Foods for Children: A Literature Review

**DOI:** 10.3390/ijerph19084756

**Published:** 2022-04-14

**Authors:** Amanda Rodrigues Amorim Adegboye

**Affiliations:** 1School of Nursing, Midwifery and Health, Faculty of Health and Life Sciences, Coventry University, Priory Street, Coventry CV1 5FB, UK; a.adegboye@coventry.ac.uk; 2Centre for Healthcare Research, Coventry University, Priory Street, Coventry CV1 5FB, UK

**Keywords:** complementary food, school meal, children, edible insects, novel food formulations, malnutrition, nutrition deficiency

## Abstract

Background: Childhood malnutrition is an important public health problem. Animal protein provides essential amino acids in a more adequate pattern than plant-based protein. However, the production of sufficient animal-sourced protein to feed the growing world population is a serious challenge. This review aims to explore the evidence on the use of edible insects as an alternative source of protein and micronutrients in complementary foods for children and their potential to address childhood malnutrition. Methods: Searches were conducted in two electronic databases PubMed and Cochrane. The reference lists of included studies were also searched. Results: Twelve studies were included in this review. All insect-enriched formulations (e.g., biscuits, cereals, porridge, paste, etc.) exceeded the daily recommended amount of protein and fat for children’s complementary foods and showed good acceptability. Only two studies assessed the efficacy of insect-enriched foods on nutritional indicators and found no effect on the reduction of stunting and wasting. However, one study found improvements in the haemoglobin levels and fewer cases of anaemia in the intervention group. Conclusions: Insect-enriched complementary foods for children are safe, acceptable and have the potential to tackle micronutrient deficiencies. More studies are needed to examine their effect on nutritional status in children.

## 1. Introduction

Access to nutritionally adequate and safe foods is part of basic human rights [1]. Access to adequate nutrition plays a critical role in the health and wellbeing status of an individual [2]. It also improves cognition and school attainment, enhances productivity and income, and halts the cycle of poverty [3]. Adolescent girls who are well-nourished and healthy are more likely to stay in school longer, have better school performance and delay their first pregnancy [3]. Well-nourished women are more likely to have fewer complications during pregnancy and deliver healthier babies who consequently are more likely to have better health outcomes later in life [4]. Infants and children who are well-nourished have a stronger immune system and are less prone to infection and diseases [4,5].

Despite the social and health benefits of good nutrition, particularly during key stages of the life cycle (e.g., pregnancy and childhood), nearly two in three children aged 6–24 months (weaning period) are not fed the minimum diet they need to meet the required levels of essential nutrients. This increases their risk of suffering from malnutrition, infections, poor brain development and impaired learning [6].

Even prior to the COVID-19 pandemic, the statistics on childhood malnutrition were alarming and the world was off-track in achieving the 2025 World Health Assembly nutrition targets and ending hunger by 2030 [7]. In 2020, 149 million children under five years (1 in 5 children) were stunted as a result of malnutrition and infections and 45 million were wasted. Approximately, one-quarter of children suffering from waste lived in sub-Saharan Africa and over half lived in South Asia [7]. Micronutrient deficiencies and overweight were also of global concern. In 2020, approximately 5.7% of children under five years were affected by overweight and 40% had anaemia [7]. It is worth noting that children suffering from overweight can be concomitantly affected by micronutrient deficiencies [8]. These conditions characterise the triple-burden of malnutrition including undernutrition, in the form of stunting and wasting; micronutrient deficiencies; and a growing prevalence of overweight and obesity [8].

The COVID-19 pandemic has significantly exacerbated household food insecurity and overturned the economic growth of many countries across the globe. These factors could hinder reductions in undernutrition prevalence, particularly in low-income and middle-income countries [9]. There is an urgent need to identify sustainable solutions to address childhood malnutrition and hunger. The publication of the FAO pioneering report on “Edible insects: future prospects for food and feed security” in 2013 [10] has generated a debate on whether edible insects could help to address food insecurity from a global perspective [11]. Edible insects have a high reproduction rate, short life cycle and thrive at densities making them ideal for vertical rearing. They do not require significant land space and water and produce less greenhouse gas compared to traditional livestock [10]. Although their nutrient composition varies across species, edible insects are, in general, a good source of protein, lipids, fibre, iron, zinc and calcium [11].

There are over 1900 species of edible insects and they are part of the traditional diets of nearly 2 billion people in the world [10]. However, in some cultures, edible insects are not well accepted as food [12]. Therefore, previous literature has focused on the potential use of edible insects as an alternative source of protein for animal feed and only recently attention has been given to the use of insects as food [10]. Some emerging studies have investigated consumers’ acceptance of food products (bread, pasta, biscuits, etc.) enriched with insect powder [13,14,15,16]. However, only a few randomised clinical trials have investigated the impact of edible insects as a dietary supplement on health-related outcomes [17].

A randomised clinical trial (RCT) has explored the use of edible insects as a dietary protein supplement to improve muscle mass and strength during resistance training in 18 young men and did not find any significant improvements in body composition and muscle strength [18]. The authors argued that a high habitual protein intake in both the control and experimental groups may explain the non-superior effect of insect protein supplementation [18]. Another small double-blind, randomised crossover clinical trial investigated whether 25 g of cricket powder/day increased the number of beneficial gut bacteria in 20 healthy adults [19]. The study found that consumption of 25 g of cricket powder per day was safe and improved gut health and reduced systemic inflammation [19]. Despite preliminary evidence of the beneficial effect of insect consumption on human health, more studies are required to fully understand its underlying biological mechanisms.

This narrative review aims to explore some of the existing evidence on the use of edible insects as an alternative source of protein and micronutrients for complementary food for infants and children and its potential impact on the indicators of childhood malnutrition, particularly in countries where edible insects are part of traditional diets.

## 2. Materials and Methods

In this study, a narrative literature review was conducted to summarise the evidence on the feasibility of using edible insects to enrich complementary foods for children and the impact of these formulations on children’s nutritional status. It considered studies evaluating the nutritional composition, safety, and acceptance of novel food products enriched with edible insects especially developed for infants’ and/or children’s consumption.

### 2.1. Literature Search

A literature search was conducted using PubMed-NCBI and Cochrane Library databases from inception to 2022 to find relevant articles using several search terms: (((infant * OR Child *) OR (children[MeSH Terms])) AND (insect * OR mealworm OR maggot * OR “Black soldier Fly” OR grasshopper* OR locust* OR Caterpillar*)) AND (malnutrition OR nutrition * OR diet * OR meal OR food * OR growth OR weight). The search was restricted to the titles, abstracts and keywords of all indexed articles. In the PubMed database, a filter was also applied to restrict the search to human studies.

The reference list of included studies was also scrutinised to identify any potentially relevant studies. Retrieved records were uploaded into Ryyan [20] for screening and full-text selection performed by one reviewer based on the pre-specified eligibility criteria described below.

### 2.2. Inclusion and Exclusion Criteria

Only studies reporting the acceptability, nutritional value, safety and effectiveness of insect-based foods, developed specifically for infant complementary feeding or for children’s school meals or supplementary feeding were eligible. There was no restriction regarding study design, duration of the study, inclusion rate of insects in the formulations, type of insect species and year of publication. Studies published in languages other than English, Portuguese and Spanish were excluded.

### 2.3. Study Selection

The search retrieved 534 records. After title and abstract screening, 14 records were selected for full-text assessment. Based on the eligibility criteria, 3 records were excluded after full-text assessment. Of the 11 selected records, two [21,22] (conference abstract and full-text paper) reported data from the same study. The full-text paper [22] was used as the main reference and it was counted as one record. The reference lists of the remaining 10 eligible papers were scrutinised and 2 additional records were identified. Thus, this review included 12 studies in total. The selection of eligible studies is described in Figure 1.

## 3. Results

### 3.1. General Characteristics of Included Studies

The general characteristics of the studies included in this narrative review are described in Table 1 and Table 2. Ten studies were conducted in Africa [23,24,25,26,27,28,29,30,31,32] and two studies in Asia [22,33]. The most frequent country was Kenya with five studies [24,27,28,29,30]. No study was conducted in Western countries. The insect species mostly represented was cricket (41.7%) [22,24,27,28,29] followed by caterpillar (16.7%) [25,26].

Two of the included studies were ongoing trials without available published findings [24,29] and two studies developed an insect-based food formulation but did not perform sensory analysis or test its efficacy on the nutritional status of infants or children [23,32]. The sample sizes of the included studies ranged from 20 [25] to 360 children [22] aged from 6 months to 10 years. Two studies [25,26] recruited infants up to 12 months of age (Table 2).

### 3.2. Nutrient Profile

The substitution rates of insects in the food formulations varied from 5% [23,33] to 30% [23,32]. In four of the trials, insects substituted less than 10% of conventional flour or protein source (e.g., milk). One study in Cambodia offered a pre-packed sachet (41 g) with 100% cricket powder as a daily ration [22]. Two studies did not provide information about insect inclusion rates [25,26]. The most frequent preparation was insect-enriched cereal/porridge [25,26,28,31], followed by insect-enriched biscuits [27,33] and insect flour [22,30].

Table 3 shows the macro and micronutrient composition of the insect-based formulations and the recommended values for the nutrient composition of fortified processed complementary foods proposed by Lutter and Dewey [34]. All insect-based formulations exceeded the daily recommended amount of protein and fat for complementary foods for 6 to 23 months old children. However, some of the food formulations did not meet the requirement for energy [22,25,26,31].

Parker et al. [32] found that 32 g of palm weevil larvae paste met 99% and 84% of recommended daily allowance (RDA) for 6–12 months old infants and 1–3 years of children, respectively. However, palm weevil larvae alone did not provide a complete amino acid profile. They offered adequate quantities of four essential amino acids (histidine, isoleucine, phenylalanine, and valine) but insufficient quantities of the remaining five essential amino acids (e.g., leucine, lysine, methionine, threonine, tryptophan). Therefore, the palm weevil larvae were mixed with peanut paste and canola oil to enhance the amino acid and lipid profile of the formulation [32].

Dewi et al. [33] reported that 60 g of 10% grasshopper-enriched biscuits contributed as much as 38% RDA to the adequacy of protein for children aged 12–24 months, compared to 24% RDA of control biscuits. The authors also reported that even with a substitution rate of 5%, grasshopper-enriched biscuits can meet the adequacy of the amino acids leucine (123%), lysine (65.6%), phenylalanine (80.8%), and methionine (134.3%) in children aged 12–24 months.

Only five studies [23,28,30,31,32] provided data on calcium content and all formulations were below the requirements of 100–200 mg. Most of the formulations did not meet the requirements for iron and zinc and the overall content of these minerals varied considerably across formulations. The grasshopper- and locust-enriched complementary foods were particularly low in iron and zinc, while honeybee larvae and termite-based formulations had the highest content of iron per serving (approximately 20 mg/serving). Mekuria et al. [31] reported that the honeybee larva formulation had the highest content (per 100 g) of iron (40.9 mg/100 g) and calcium (68.2 mg/100 g) compared to the plant-based and commercial formulations.

### 3.3. Acceptability of Insect-Based Formulations

Four studies did not provide data on the acceptability of the formulations [23,24,29,32]. Three studies assessed the acceptability of the food product formulations of both mothers or caregivers and children [25,28,30] and three studies only assessed the children’s acceptability [22,26,27]. The remaining studies assessed product acceptability on a sample of semi-trained mothers [31] or panellists [33].

The acceptability summary scores (mean or median) for the sensory attributes of the formulations were presented in Table 4. Sensory attributes (e.g., taste, smell, colour, etc.) were assessed using a hedonic test and Likert-style response scales. In most of the studies, children’s acceptability was measured as the total amount of food consumed by the children during the study period. The threshold for acceptance varied from >50% to ≥75% of children’s consumption of the serving provided.

In general, studies reported good acceptability of the new formulations. For example, Bauserman et al. reported that all mothers liked or liked very much the taste and the overall impression of the caterpillar cereal/porridge [25].

There is evidence that the level of acceptability was related to the insect substitution rate in the formulations. Dewi et al. [33] found that the acceptability of sensory attributes (taste, colour, aroma, and texture) of grasshopper biscuits decreased with an increase in the level of grasshopper flour substitution (5%, 7% and 10%) and the most preferred grasshopper biscuit was the one with 5% flour substitution.

It was observed that the sensory acceptability of the formulations increased over time. Homann et al. [27] found that milk-based biscuits had high ratings in week 1 and the ratings continued to rise until week 4. Cricket-based biscuits followed the same trend but at lower ratings. Kinyuru et al. [28] reported an increase in the proportion of children who consumed >75% of the cricket porridge serving from 55% in week 1 to 70% in week 4. A similar pattern of increased children’s consumption over time was also observed in other studies [22,30].

It was noted the insect-enriched biscuits were darker in colour compared to the control biscuits [27,33]. Homann et al. [27] reported that milk-based and cricket-based biscuits had different slopes for colour and smell rating over time, suggesting a slower adaption for these two sensory properties in novel insect-based biscuits compared to conventional biscuits.

No serious adverse events were reported [25,27,30]. However, Bauserman et al. [26] reported one single episode of vomiting on the first day of the trial without any further events during the study period.

### 3.4. Microbiological Safety

The description of the microbiological profile of the formulations was not described in all studies. Mekuria et al. [31] reported that microbial counts (CFU/g) of *E. Coli*, *S aureus*, *Salmonella*, *Shigella*, yeast, mould and total plate count were below the acceptable level [35] in the bee larva-based formulation at 3 and 6 months, indicating a good shelf-life of the product. Similar results were reported for 10% edible termite flour [30], caterpillar cereal [25] and 10% cricket-based biscuits [27].

### 3.5. Health and Nutritional Status

Only two trials provided data on nutritional status outcomes and the key findings are presented in Table 5 [22,26]. A cluster randomised trial conducted in the DRC provided caterpillar cereal/porridge combined with nutrition education to 6-month-old infants until they were 18 months old [26]. The study did not have any significant effect on the prevalence of stunting and wasting even though the insect-based meal provided an extra 132–198 kcal for 12 months. Infants in the intervention group had higher haemoglobin concentrations and a lower prevalence of anaemia compared to infants in the control group at 18 months [26]. The study did not find any significant difference in the incidence of infectious diseases between the two groups. However, weekly symptom recall was lower in the intervention group (44% vs. 66%) compared to the control group but the difference was not statistically significant (*p* = 0·22) [26]. Similar findings were observed in a cluster randomised trial in Cambodia [22]. The study consisted of three intervention groups including moringa powder and cricket powder combined with nutritional education and counselling (CEN) and CEN alone. The provision of moringa and cricket did not improve the nutritional status of children, though consumption of these foods helped children to meet their energy, iron, and zinc requirements [22].

## 4. Discussion

This review was based on 12 studies, which consistently showed the feasibility of enriching complementary foods for infants and children with edible insects. All formulations met the protein and fat requirements for complementary food for 6–23-month-old children, but not all formulations fulfilled the energy and micronutrient requirements [34].

Bauserman et al. [25] stated that a serving of 30 g of caterpillar cereal offered as a sole complementary food is unlikely to meet the energy requirements for infants. However, the formulation might be considered a satisfactory supplement to breast milk and existing complementary foods which provide adequate energy from carbohydrates. It is worth noting that the nutritional composition of the complementary foods depends on the portion size and the type and quantity of other ingredients included in the formulations (e.g., nuts, herbs, grains, legumes, oils, etc.). The use of insect-enriched complementary foods could be an alternative to increase nutritional intake and promote dietary diversification.

Although the literature shows that the house cricket (*Acheta domesticus)* contains all nine essential amino acids [36], none of the studies proposing a cricket-enriched formulation provided a detailed analysis of the amino acid profile of the formulations and whether they met the dietary recommendations. On the other hand, some edible insect species (e.g., palm weevil larva) do not provide a complete amino acid profile. However, they offer a significant amount of essential nutrients and can be feasibly integrated into agriculture and nutrition interventions to tackle upstream causes of malnutrition [32].

The overall iron content varied significantly across formulations and 15 of the proposed formulations did not meet the minimum requirement of 7–11 mg per serving [34]. Despite the low amount of iron in some formulations, the literature suggests that the iron bioavailability of edible insects is comparable to beef. An in vitro study has assessed iron bioavailability from beef, edible insects, and durum whole-wheat flour and found that cricket and beef had similarly higher levels of iron than grasshopper, mealworm, and buffalo worms. However, the iron solubility was significantly higher in the insect samples than in beef [37]. Combining whole-wheat flour with insect or beef protein resulted in a decrease in the mineral content and iron solubility of the mixture. The study also showed that grasshopper, cricket, and mealworms contain significantly higher chemically available calcium, copper, magnesium, manganese, and zinc than beef [37]. Bauserman et al. [26] observed a statistically significant increase in the haemoglobin concentrations and a decrease in the prevalence of anaemia among children consuming caterpillar cereal/porridge for 12 months compared to the control group.

Edible insects also have a high content of fibre. The main source of fibre is chitin which comes from their hard exoskeleton. The literature suggests that chitin could reduce the bioavailability of nutrients and protein digestibility [38]. Antinutrient content of formulations could also reduce the bioavailability of nutrients including iron, zinc and calcium and consequently have a negative impact on the children’s nutritional status. Mekuria et al. [31] reported higher levels of tannins and phytates (mg/100 g) in the plant-based porridge (208.9 mg and 68.2 mg) compared to bee larva-based porridge (119.4 mg and 13.1 mg).

The use of the extrusion cooking method has several beneficial effects when developing the formulations of complementary foods for children. It can limit antinutritional factors and microorganism contamination [31,39]. The microbial load of extruded products is a good indicator of product safety and shelf life [39]. The studies showed that the microbial load of formulations was low and deemed safe for human consumption [25,27,30,31]. No study reported any serious adverse event associated with insect consumption.

Although formulations were considered safe for human consumption, data on the effectiveness of these formulations in addressing malnutrition were limited. Only two studies investigated the impact of insect consumption on children’s nutritional status and found no statistically significant differences in the anthropometric indicators between the groups [22,26]. One study found a statistically significant improvement in the haemoglobin levels and a reduction in the prevalence of anaemia in the intervention group [26]. The other study observed improvements in the haemoglobin and ferritin levels in all groups [22]. However, this study did not have a control group and all comparison groups received some type of intervention (cricket + CEN, moringa CEN and CEN alone) [22]. Furthermore, some of the baseline characteristics were unbalanced among groups. The authors highlighted that the amount of cricket offered and consumed by the children might not have been sufficient to make significant improvements in the children’s nutritional status [22]. The second study experienced a high loss of follow up [26]. Although there were no significant differences between dropouts and completers, the final sample size might have been insufficient to detect modest differences in nutritional status indicators between the intervention and control group [26].

Overall, the studies showed good acceptability of the food formulations. However, sensory acceptability scores for insect-enriched foods were lower than the scores for traditional or control foods. Mekuria et al. [31] stated that the high score rating of the commercial wean mix could be due to the addition of flavouring to the product. Studies also showed that both sensory acceptability scores and consumption increased over time [22,28,30]. This suggests that children’s consumption of less familiar food can increase with multiple exposures, and they might require a longer time to adapt to the sensory properties of the novel insect-based formulations. This finding is in line with previous literature on children’s acceptance of different types of novel foods (e.g., vegetables and snack bars) which has shown that repeated exposure is a critical factor for promoting consumption of novel foods among children [40,41,42].

### 4.1. Limitations

This is a narrative review and therefore not all available evidence on the topic was considered. However, the two main health-related electronic databases were included in the search and the reference lists of eligible papers were scrutinised. Furthermore, some elements of the systematic review methodology were applied to minimise potential bias. There are more than 1900 species of edible insects in the world, but the search considered only the most commonly reported species in the literature. There is a chance that studies including less popular edible insects were not included in this review. Additionally, only two trials contributed data for the impact of edible insects on nutritional outcomes.

### 4.2. Implications for Future Research

Even though entomophagy (the practice of eating insects) is common in countries in Africa, Asia and Latin America, there are few observational and experimental studies on the nutrition and health outcomes associated with edible insect consumption [11]. In this review, 17 different insect-based formulations were identified and most of the formulations did not meet the micronutrient requirements for complementary feeding. This could potentially be addressed by increasing the substitution rate of some formulations. However, higher substitution rates are likely to alter the colour and general appearance of the novel food and potentially decrease their acceptability.

Not only the quantity but the quality of micronutrients should be considered during the formulation process. Therefore, future studies should assess the bioavailability of micronutrients during the normal shelf life of the novel product and potential interactions with flavour or colour systems.

In high-income countries, commercially fortified complementary foods (e.g., baby porridges) are commonly consumed, but they are often unaffordable in low-income countries and deprived communities. The literature shows that homemade complementary foods remain commonly used in African countries [43]. However, unfortified plant-based complementary foods including those based on improved recipes do not provide sufficient micronutrients particularly, iron, zinc, and calcium for children [43]. In this context, where insects are easily available, strategies to promote their incorporation into traditional complementary food recipes as a direct nutrition intervention could be potentially cost-effective, especially when it can replace synthetic vitamin and mineral fortification. However, the use of edible insects to address malnutrition is an emerging topic and only two clinical trials were included in this review [22,26]. The studies did not find any significant effect of insect-enriched formulations on children’s anthropometric indicators and did not provide information about the cost of the interventions and their feasibility to be implemented at scale (at a local, regional or national level). Therefore, additional large-scale and well-designed clinical trials are still required to investigate the ideal formulation including portion size, insect species, and insect substitution rate, and the adequate timing, duration and intensity of interventions to optimise their impact on nutrition and health outcomes in children. It is also important to assess whether edible insects, as an alternative source of protein, fat and micronutrients, have the potential to provide cultural, environmental and economic benefits to communities affected by malnutrition.

## 5. Conclusions

The literature suggests that insect-enriched complementary foods for children are safe and acceptable. Given adequate formulations (e.g., substitution rate) and appropriate children’s consumption, insect-enriched complementary foods have the potential to tackle iron deficiency, especially in countries most affected by malnutrition and food insecurity and where consumption of edible insects is already part of the diet.

It is important to consider the cultural adequacy dimension of the right to food. Food should be seen beyond its nutrient content as it is a critical part of human social and cultural functions. The selection of edible insects which are locally available, and part of the traditional diet can improve the acceptability of formulations while addressing nutrition deficiencies. In this context, governments could encourage the consumption and production of locally available insects, particularly in communities highly affected by malnutrition [11].

## Figures and Tables

**Figure 1 ijerph-19-04756-f001:**
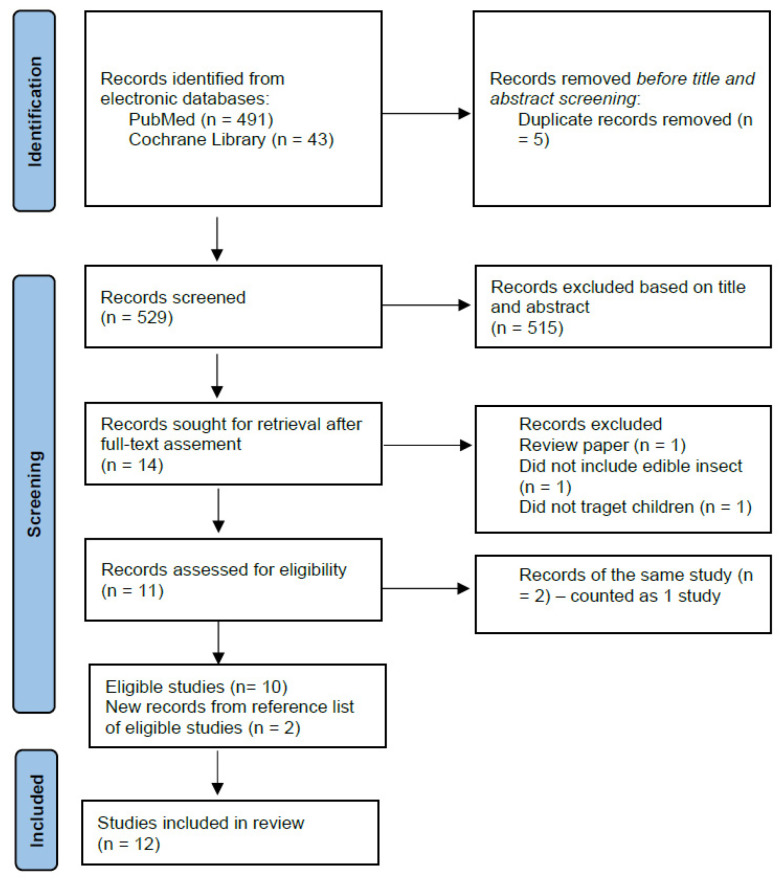
PRISMA flow diagram of selection of eligible studies.

**Table 1 ijerph-19-04756-t001:** General characteristics of all included studies.

Author, Year	Country/Study Name	Insect Species	Study Aim
Akande et al., 2022 [23]	Nigeria	Migratory locust (*Locusta migratoria*)	To explore the feasibility of replacing powdered milk with locust powder as an affordable and sustainable ingredient in ready-to-use therapeutic foods (RUTF) for the treatment of malnutrition.
Boonstra, et al., 2021 [24]	Kenya Zaza study	House cricket (*Acheta domesticus*)	To estimate zinc absorption from dried crickets added to non-refined maize porridge in comparison with a plain porridge in children.
Bauserman et al., 2015 [25]	Democratic Republic of Congo (DRC)	Caterpillar	To develop a cereal enriched with caterpillars and other locally available ingredients as a potential complementary feeding product for infants, and to assess its nutritional content and microbiologic contamination.
Bauserman et al., 2015 [26]	Democratic Republic of Congo (DRC)	Caterpillar	To evaluate the efficacy of a caterpillar-based cereal, as a micronutrient-rich, locally available alternative animal-source food, on decreasing the prevalence of stunting and anaemia in infants in the DRC.
Dewi et al., 2020 [33]	Indonesia	Wood grasshopper (*Melanoplus cinereus*)	To evaluate the nutritional composition and organoleptic properties of the new baby biscuit formulation with grasshopper flour.
Homann et al., 2017 [27]	Kenya	House cricket (*Acheta domesticus*)	To develop a cricket-based biscuit formulation suitable for school feeding programmes and assess its acceptability among schoolchildren in Kenya, in comparison to a milk-based biscuit.
Kinyuru et al., 2021 [28]	Kenya	Cricket	To develop a formulation for a cereal-cricket porridge suitable for Kenyan school feeding programmes, and to evaluate its safety acceptability in comparison to cereal and cereal-milk porridges.
Kipkoech & Ross, 2017 [29]	Kenya	House cricket (*Acheta domesticus*)	To assess the effect of edible cricket in a school feeding programme on nutritional status, gut microbiota and health in pre-school children in Kenya.
Konyole et al., 2012 [30]	Kenya	Termites (*Macrotermes subhylanus*)	To evaluate the acceptability of two types of flours and porridges as complementary foods for young children containing germinated grain amaranth and maize with and without edible termites and *dagaa* fish.
Mekuria et al., 2021 [31]	Ethiopia	Honeybee larvae (*Apis mellifera*)	To assess the nutrient profile, microbial safety, and sensory characteristics of complementary foods based on blends of staple grains and honeybee larvae for infants aged 6–12 months.
Menasria et al., 2018 [22]	Cambodia	Cricket	To evaluate the efficacy of two local foods in combination with nutritional education and counselling (CEN) compared to CEN alone on improving nutritional status and dietary intake of children.
Parker et al., 2020 [32]	Ghana	Palm weevil larvae (*Rhynchophorus phoenicis*)	To evaluate the nutrient profile of palm weevil larvae also known as akokono, and to assess the potential of palm weevil larvae as an ingredient in complementary foods for Ghanian children.

**Table 2 ijerph-19-04756-t002:** Description of study design, intervention groups, and formulations.

Author, Year	Study Design	Study Population	Intervention Groups and Formulations	Dose and Duration
Akande et al., 2022 [23]	Food product development	N/A	Seven types of RUTFs were formulated. Milk powder was replaced by locust powder at 5–30% levels. Ingredients: milk powder, locust powder, sugar, peanut butter, vegetable oil, mineral and vitamin premix	92 g sachet
Boonstra, et al., 2021 [24]	Randomised crossover trial	32 ^a^ healthy children aged 24–36 months	Meal test 1—porridge with 6% of non-refined maize flour combined with 15–20 g of freeze-dried 67 Zn intrinsically labelled crickets Meal test 2—porridge with 7–8% of non-refined maize combined with protein powder extracted from crickets (15–20 g 67 Zn intrinsically labelled crickets) Meal test 3—porridge with 7–8% of non-refined maize combined with ZnSO4 labelled with 0.75 mg of 68 Zn and 2.25 mg of unlabeled Zn added as dissolved ZnSO4. Meal test 4—porridge with 7–8% of non-refined maize and 0.75 mg of labelled 68 Zn added as dissolved ZnSO4 Each test meal contains ~3 mg of zinc (isotopic enrichment 25%)	One meal (200 g of porridge) on one test day One test day for each meal
Bauserman et al., 2015 [25]	Experimental study without control	20 infants aged 8–10 months old and their mothers	Caterpillar cereal meal: ground corn, palm oil, sugar, salt and caterpillar flour 1:1 ratio of caterpillar flour to cornflour	7 sachets with 30 g of cereal mixed with 100 mL of boiling water offered to the infants 3 times during the week
Bauserman et al., 2015 [26]	Cluster randomised controlled trial	175 infants aged 6 months old	Caterpillar cereal meal: ground corn, palm oil, sugar, salt, caterpillar flour Usual diet	Daily portion of pre-packaged caterpillar cereal from 6 months of age until 18 months of age Daily portion of 30 g and 45 g for infants 6–12 months and 12–18 months of age, respectively
Dewi et al., 2020 [33]	Experimental study (sensory evaluation of a complete randomised one factor by replacement of grasshopper flour contents)	25 semi-trained panellists	36.4% wheat flour biscuit (control) 5% grasshopper and 31.4% wheat flour biscuit 7% grasshopper and 29.4% wheat flour biscuit 10% grasshopper and 26.4% wheat flour biscuit Ingredients: margarine, sugar, skimmed milk powder, egg yolk, wheat flour and grasshopper flour.	Serving size for sensory test: 12 g of each of the four types of biscuits.The suggested serving size for infants: 6 pieces (60 g)
Homann et al., 2017 [27]	Randomised parallel intervention	54 schoolchildren aged 5–10 years	10% of cricket-based biscuit 10% of milk-based biscuit	100 g of biscuits containing either 10% of cricket powder or 10% of milk powder during school days for four weeks
Kinyuru et al., 2021 [28]	Randomised parallel intervention	138 schoolchildren aged 3–5 years and 73 caregivers	MM Porridge: 60% maize and 25% millet and 10% oil M10 Porridge: 50% maize, 25% millet and 10% milk powder and 10% oil C5 Porridge: 55% maize, 25% millet, 5% cricket powder and 10% oil All porridges were fortified with vitamin and mineral premix (1.8 g/100 g flour formulation)	One serving of 300 mL porridge (65 g porridge flour) during school days for four weeks
Kipkoech & Ross, 2017 [29]	Randomised clinical trial	134 ^a^ schoolchildren aged 3–4 years	Intervention: porridge enriched with cricket Positive control: similarly enriched porridge with milk powder Negative control: a fortified plant-based porridge	Provision 65 g of either milk-based, cricket-based or cereal-based flour in form of porridge five days per week (Monday to Friday) as part of the school meal for 6 months
Konyole et al., 2012 [30]	Randomised crossover trial	57 children aged 6–24 months and their mothers	Winfoods Lite (WFL) porridge: germinated grain amaranth, maize soy oil and sugar Winfoods Classic (WFC) porridge: 3% *dagaa* and 10% edible termites added to WFL Corn-soy blend (CSB+) porridge	One serving of 150 mL of porridge One test day for each meal with one-day washout between meals
Mekuria et al., 2021 [31]	Experimental study (sensory evaluation)	30 semi-trained mothers	Complementary foods (ComF1) 1: 57% white maize, 29% red teff and 14% soybean ComF2: 58% white maize, 29% red teff and 13% insect bee larvae Commercially available wean mix	One serving (50 g of flour to 250 mL of water) of each complementary food on one test day
Menasria et al., 2018 [22]	Cluster randomised controlled trial	360 children aged 6–23 months	Nutritional education and counselling (CEN) alone CEN plus moringa powder CEN plus cricket powder	Daily ration of 16 g of moringa and 41 g of cricket powder for 6 months
Parker et al., 2020 [32]	Food product development	N/A	Palm weevil larva-peanut paste. Ingredients: dry-roasted palm weevil larvae (30%) peanuts (70%) and canola oil (2 mL oil per 100 g paste)	One serving size of two tablespoons (32 g)

^a^ anticipated target sample size of ongoing clinical trials.

**Table 3 ijerph-19-04756-t003:** The nutritional profile of one serving of insect-based meal and the referenced amounts of macronutrients in complementary foods for 6–23-month-old children.

Study	Rec	[25,26]	[26]	[31]	[32]	[30]	[33]			[22]	[28]	[27]	[23]					
Insect		Caterpillar		Bee Larva	Palm Weevil Larva	Termite	Grasshopper			Cricket			Migratory Locust					
Inclusion rate		N/A	N/A	13%	30%	10%	5%	7%	10%	100%	5%	10%	5%	10%	15%	20%	25%	30%
Meal type		C/P	C/P	C/P	Pt	Po ^a^	B			Po	C/P	B	RUTF					
Portion size (g)	50	30	45	50	32	100	60			41	65	100	92					
Protein (g)	3–5.5	6.9	10.3	5.9	10.9	21.5	8.6	9.2	9.9	23	6	13.9	18.6	18	19.9	19.2	24.3	21.9
Carbohydrate (g)	N/A	12	18	31.4	5.4	N/A	34.2	32.8	31.3	N/A	40.7	59	37.4	37.5	36.3	38	34	35.1
Fat (g)	6.3	6.3	9.4	7.2	14.8	18.5	11.8	13.2	13.3	N/A	9.2	19	29.3	28.7	27.9	27.9	26.3	27.2
Kcal	220	132	198	213.6	N/A	539.7	227	287	284	196	251.4	462.6	487.8	480.2	476.6	480	469.4	473
Iron (mg)	7–11	3.8	5.7	20.5	0.48	20.2	0.24	0.22	0.24	2.6	5.6	1.6	0.06	0.09	0.08	0.06	0.09	0.05
Zinc (mg)	4–5	3.8	5.6	1.5	N/A	5.1	0.05	0.05	0.06	7.2	N/A	3.1	0.07	0.09	0.07	0.05	0.06	0.08
Calcium (mg)	100–200	N/A	N/A	22.2	13.73	24.7	N/A	N/A	N/A	N/A	39.1	N/A	0.9	1.2	0.9	1.1	0.7	0.7

Values were extracted from the original manuscript and rounded. ^a^ author did not provide information about the amount of powder required to prepare a serving of 150 mL porridge. Values refer to 100 g of powder/flour including 10% edible termites and 3% *dagaa* and 10% edible termites. B—biscuit, C/P—cereal or porridge, N/A—data not available, Po—powder, Pt—paste, RUTF—ready-to-use therapeutic food. Rec—the recommended amount of macronutrients in complementary foods for 6–23-month-old infants proposed by Lutter and Dewey [34].

**Table 4 ijerph-19-04756-t004:** Summary of overall acceptance or level of consumption of novel insect-enriched meal as complementary feeding for children.

Author, Year	Domains Assessed	Type of Data and Response Scale	Key Results
Akande et al., 2022 [23]	N/A	N/A	N/A
Boonstra, et al., 2021 [24]	N/A ^a^	N/A	N/A
Bauserman et al., 2015 [25]	Mothers’ acceptance Hedonic test: smell, taste, texture, colour, consistency and overall impression Infants’ acceptance Consumption of ≥75% of the cereal in the last 4 days of the trial and no adverse symptoms related to cereal consumption.	5-point Response scale (5 = like very much)	Mothers’ median ratings: overall impression = 4, taste = 5, smell = 4, texture = 4, colour = 5, and consistency = 4 All infants consumed ≥75% of the daily cereal portion, 26% consumed 100% of the cereal. 1 infant experienced 1 episode of vomiting. No severe adverse event was reported
Bauserman et al., 2015 [26]	Infants’ acceptance Consumption of > 70% of the serving	Cereal consumption: collection of unused food sachets and maternal report on infant consumption	Infant consuming >70% of the serving = 90% % infants consuming >90% of the serving = 62%
Dewi et al., 2020 [33]	Panellist acceptance Hedonic test: colour, texture, smell and taste of 0%, 5%, 7% and 10% grasshopper flour	4-point Response scale (4 = dislike extremely)	Panel mean rating: 0% grasshopper flour taste = 1.4, colour = 1.4, smell = 1.5 and texture = 1.6 5% grasshopper flour taste = 2.1, colour = 2.2, smell = 2.5 and texture = 2.1 7% grasshopper flour taste = 2.4, colour = 2, smell = 2.7 and texture = 2.4 10% grasshopper flour taste = 3.1, colour = 2.5, smell = 2.8 and texture = 2.4
Homann et al., 2017 [27]	Schoolchildren Hedonic test: appearance, smell, texture and overall Schoolchildren’s acceptance Biscuit consumption: average acceptance > 50%, good acceptance > 75% and long-term good acceptance > 75% for at least 75% of the study days	5-point Response scale with smileys (5 = like a lot) Weight of biscuits eaten and reluctance and rejection to eat were recorded	Range of ratings during 4 weeks of the experiment Cricket-based biscuit overall = 5–2.5, taste = 5–2.5, smell = 5- 2, texture = 5–2, colour = 5–3 and appearance = 5–4 Milk-based biscuit overall = 5–4, taste = 5–4 smell = 5–4, texture = 5–4, colour = 5- 3 and appearance = 5 Overall children’s consumptions were 96.9% and 94.2% for cricket-based and milk-based biscuits
Kinyuru et al., 2021 [28]	Caregivers’ acceptance Hedonic test: smell, taste, colour and overall Children’s acceptance Porridge consumption: >75% highly acceptable, 50–75% moderately acceptable and <50% least acceptable	7-point Response scale (5 = like very much) Total amount of porridge consumed	Caregivers most preferred M10 porridge colour (6.4) and taste (5.5). Overall, all the porridges (MM, M10 and C5) recorded overall acceptability scores of ≥5 % children consuming >75% of the serving at the end of week 4: MM = 100% M10 = 100% C5 = 70% 5% of the children recorded <50% acceptance of C5 porridge in week 4
Kipkoech & Ross, 2017 [29]	N/A ^a^	N/A	N/A
Konyole et al., 2012 [30]	Mothers’ acceptance Hedonic test: smell, texture and colour Infants’ acceptance Consumption of at least 75% of the porridge serving as acceptable	5-point Response scale (5 = like very much) The amount of porridge consumed was calculated by the difference between the total amount provided and the amount left and spilt	Mothers’ mean ratings: WFL smell = 3.9, texture = 4.6 and colour = 3.9WFC smell = 3, texture = 4.4 and colour = 3.3CSB+ smell = 3.8, texture = 2.4 and colour = 3.8 % infant consuming at least 75% of served porridge: WFL = 43% WFC = 19.6% CSB+ = 21% No adverse events were reported for all the foods during the study.
Mekuria et al., 2021 [31]	Mothers’ acceptance Hedonic test: appearance, smell, taste, texture and overall	5-point Response scale (5 = like very much)	Complementary food 1 (soy-based) Appearance = 3.8, smell = 3.7, taste = 3.6, texture = 3.8 and overall = 3.6 Complementary food 2 (bee larvae-based) Appearance = 4.1, smell = 4.2, taste = 4.4, texture = 4 and overall = 4.2 Commercial wean mix Appearance = 4.4, smell = 4.4, taste = 4.5, texture = 4.5 and overall = 4.6
Menasria et al., 2018 [22]	Infants’ acceptance Consumption of >50% of the daily ration of moringa (16 g) and cricket (41 g)	Consumption of cricket and moringa assessed from three 24 recalls (baseline, midterm and endline)	% infant consuming >50% of the daily ration: Moringa powder = >60% (midterm) and 100% (endline) Cricket powder = 7% (midterm) and 79% (endline)
Parker et al., 2020 [32]	N/A	N/A	N/A

^a^ ongoing study. Information was extracted from the trial registration form. CSB+—corn-soy blend porridge, C5—maize, millet and cricket powder, MM—maize and millet porridge, M10—maize, millet and milk powder porridge, N/A—not available, WFL—Winfoods Lite: porridge with germinated grain amaranth, maize soy oil and sugar, WFC—Winfoods Classic: 3% *dagaa* and 10% edible termites added to WFL.

**Table 5 ijerph-19-04756-t005:** Nutritional status indicators.

Author, Year	Wasting and Stunting	Other Anthropometric Indicators	Anaemia and Hb Levels	Health Status
Bauserman et al., 2015 [26]	NS difference in stunting (67% vs. 71%, *p* = 0·69) and wasting (8% vs. 10%, *p* = 0.6) prevalence at 18 months between the caterpillar cereal/porridge and control groups	NS differences in LAZ, WAZ and linear growth velocity between caterpillar and control group at 18 months	The caterpillar cereal/porridge group had higher Hb concentration than control (10·7 vs. 10·1 g/dl, *p* = 0·03) and lower anaemia prevalence (26% vs. 50%, *p* = 0·006) at 18 months	NS difference between two groups in mortality and incidence of infectious diseases
Menasria et al., 2018 [22]	Stunting prevalence increased in the cricket group from 20.7% at baseline to 42.3% at endline, but remained unchanged in the moringa and control groups (*p* = 0.000). NS difference in wasting prevalence between baseline and endpoint in both groups	NS difference in WL/HZ between baseline and endline in the cricket and moringa groups. NS increase in the L/HZ was observed in all groups	Levels of Hb and ferritin increased in all groups including the control group between baseline and endline	% of healthy children significantly increased from baseline to endline in all groups

Hb—haemoglobin, LAZ—length-for-age Z-scores, L/HAZ—length/height-for-age Z-score, NS—no significant, WAZ—weight-for-age Z-scores, WLZ—weight-for-length Z-scores, WL/HZ—weight-for-length/height Z-score.

## Data Availability

I conducted a secondary data analysis of publicly available data.

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
