# Peer review of "Potential Use of Edible Insects in Complementary Foods for Children: A Literature Review"

_ijerph, 2022, doi:10.3390/ijerph19084756_

Round 1

Reviewer 1 Report

Thank you for the opportunity to review this important review. All manuscripts I have reviewed in the past have included line numbers. Without line numbers it is much more difficult to give detailed feedback.

I added call out comments to the manuscript I was able to download. 

Good review of a small body of knowledge, some significant editing needed, but the manuscript shows promise in bringing a viable and important nutritional option to light.

Would have liked to seen more integration in the discussion. Giving purpose to the body as a whole, what is known and what is needed. 

Conclusion seemed to overreach beyond the information discussed in the body and in the discussion. 

Author Response

REVIEWER #1

Thank you for the opportunity to review this important review. All manuscripts I have reviewed in the past have included line numbers. Without line numbers it is much more difficult to give detailed feedback.

I added call out comments to the manuscript I was able to download. 

Apologies for not adding the line numbers. I highly appreciate your annotated comments.

Good review of a small body of knowledge, some significant editing needed, but the manuscript shows promise in bringing a viable and important nutritional option to light.

Thanks for pointing out some extra spaces in the text. I have removed them. Some of the extra spacing was due to formatting or text alignment (journal template).

I have accepted all suggestions regarding wording (For example: in the introduction, I replaced the word ‘aggravated’ with ‘exacerbated’, etc.).

Thanks for pointing out poor transitions in the manuscript text. I have improved the text transition. I hope the reviewer finds the amendments satisfactory.

I did not include the word toddler as a search term, but based on the search results it seems that the word children and infants (and their variations child, infant) have retrieved relevant papers. Table 2 shows that the children’s age ranged from 6 months to 10 years.

There was no restriction regarding the year of publication. Line 99-100 states “A literature search was conducted using PubMed-NCBI and Cochrane Library databases from inception to 2022”. I have re-stated this information (lines 118) “There was no restriction regarding study design, duration of the study, inclusion rate of insects in the formulations, type of insect species and year of publication.”

As requested I have included citations number for clarity when presenting the results.

Table 1 was re-formatted to keep the words within 3 lines whenever possible.

Table 2 was re-formatted. I hope the changes have improved the readability of the table content. However, I am confident that if the paper is deemed of good quality for publication the journal will ensure that the style and formatting of the manuscript are up to the standards.

I agree with the reviewer that Table 5 is wordy. I have condensed the text. I hope the amendments have improved the readability of the table.

I agree with the reviewer that ongoing studies have not provided results data to the review. However, their inclusion in this review provides information about the study aim, study population, potential sample size, details about the intervention and outcomes of interest. Their inclusion also aids readers to keep an eye out for upcoming results. Although I acknowledge that I did not conduct a systematic review, I tried to incorporate some elements of the systematic review methodology. The Cochrane Handbook recommends that authors should strive to identify unpublished and ongoing trials to avoid publication bias.

Since the term complementary foods is used to indicate the food is complementing human milk, perhaps replacing milk is in opposition to the concept? Can you rephrase for clarity?

I tried to paraphrase the study's aim in order to preserve the original meaning. The term milk in this instance refers to milk powder as a standard ingredient in the RTUF formulation. I have amended the text as suggested to improve text clarity.

Would have liked to seen more integration in the discussion. Giving purpose to the body as a whole, what is known and what is needed. 

Thanks for this pertinent comment. The use of edible insects to address malnutrition is an emerging topic with only a few RCTs. I have added a few sentences in the discussion (under implications for future research sub-section) on what is known and what is needed as suggested. I hope you find the additional text satisfactory.

Thanks for your relevant comment in the discussion regarding the bioavailability of micronutrients. Although in the studies where nutrient analysis was presented most of the formulations did not meet the micronutrient requirements for complementary feeding. This might be due to insufficient insect substitution rates.

In high-income countries, commercially fortified complementary foods (e.g. baby porridges) are commonly consumed, but they are often unaffordable in low-income countries and deprived communities. Studies conducted in African countries showed that homemade complementary foods remain commonly used. Even when based on an improved recipe, however, unfortified plant-based complementary foods provide insufficient key micronutrients (especially, iron, zinc, and calcium) during the age of 6–23 months. In this context, where insects are easily available, especially to communities affected by malnutrition strategies to promote the incorporation of insects into traditional complementary food recipes as a direct nutrition intervention could be useful, especially when it can replace synthetic (vitamin and mineral premix) and often imported nutrient supplements / RUTF.  

Conclusion seemed to overreach beyond the information discussed in the body and in the discussion. 

Thanks for your comments. I have toned down the conclusions and amended the text (discussion section) when discussing the impact of edible insects on micronutrient deficiencies. I have amended the discussion text about the findings regarding the impact of edible insect consumption on anaemia and haemoglobin levels.  Lines 323-329 - One study found statistically significant improvement in the haemoglobin levels and a reduction in the prevalence of anaemia in the intervention group [26]. The other study observed improvements in the haemoglobin and ferritin levels in all groups [22].   However, this study did not have a control group and all comparison groups received some type of intervention (cricket + CEN, moringa CEN and CEN alone) [22]. Also, some of the baseline characteristics were unbalanced among groups. The authors highlighted that the amount of cricket offered and consumed by the children might not have been sufficient to make significant improvements in children’s nutritional status [22].

Reviewer 2 Report

The paper is in the scope of the journal. Presented study (review article) provides useful information about the potential use of edible insects in complementary foods for children and the topic is very important, however manuscript needs to be significantly improved. The most significant improvement is needed in the Method section which is not adequately presented (see comments).

There are some major and minor comments:

The Introduction section is well written and provides all important information. Some minor mistakes need to be corrected:

Instead of : “However, only a few randomised clinical trials have investigated the impact of edible as a dietary supplement on health-related outcomes [17].” should be: “However, only a few randomised clinical trials have investigated the impact of edible insects as a dietary supplement on health-related outcomes [17].

Important! Methods section needs to be improved. The literature search strategy should be presented in stepwise approach. Brackets are used to define the order in which the concepts are processed. Please use parentheses to build a search with a combination of Boolean Operators.

Results

Subchapter 3.1 Study selection should be removed from the Results chapter and should be implemented in the Methods chapter

Table 3 - change the Ref tag to another (Ref tag is misleading because it can be related to the literary sources of other papers referred to in the table - as an abbreviation of a reference). Also, the table refers to literature sources [25, 26] and [26] - is it presented correctly? Please explain.

General remarks

The font should also be uniform (e.g. see text: “The study found that consumption of 25 g cricket powder per day was safe and improved gut health and reduced systemic inflammation”)

Latin words, phrases and abbreviations, including generic and specific names, should be written in italic throughout the text (e.g. Shigella,…).

English needs a little improvement - check spelling.

Author Response

REVIEWER #2

The paper is in the scope of the journal. Presented study (review article) provides useful information about the potential use of edible insects in complementary foods for children and the topic is very important, however manuscript needs to be significantly improved. The most significant improvement is needed in the Method section which is not adequately presented (see comments).

There are some major and minor comments:

The Introduction section is well written and provides all important information. Some minor mistakes need to be corrected:

Thanks for your comment. Corrections were made as suggested (See track changes in the revised manuscript).

Instead of : “However, only a few randomised clinical trials have investigated the impact of edible as a dietary supplement on health-related outcomes [17].” should be: “However, only a few randomised clinical trials have investigated the impact of edible insects as a dietary supplement on health-related outcomes [17].

Apologies for the missing word. The text has been amended as suggested.

Important! Methods section needs to be improved. The literature search strategy should be presented in stepwise approach. Brackets are used to define the order in which the concepts are processed. Please use parentheses to build a search with a combination of Boolean Operators.

Thanks for your pertinent comment. I have included the search terms using parentheses and Boolean operators. The search included in the revised manuscript was applied in PubMed. The same search terms were used in the Cochrane library but the syntax was slightly different.

Results

Subchapter 3.1 Study selection should be removed from the Results chapter and should be implemented in the Methods chapter

Thanks for your pertinent suggestion. I have moved the sub-section on ‘Study selection’ to the methods.

Table 3 - change the Ref tag to another (Ref tag is misleading because it can be related to the literary sources of other papers referred to in the table - as an abbreviation of a reference).

I agree that using the abbreviation Ref may cause confusion. I have changed it to ‘recommended’- Rec

Also, the table refers to literature sources [25, 26] and [26] - is it presented correctly? Please explain.

Yes, the citation is correct. These are two independent studies conducted by the same group of authors. As described in Tables 1 and 2, in study 25 the authors developed a caterpillar-enriched formulation (30g portion size) and tested its acceptability in a small sample of infants and their mothers during a short period of time. In study 26, the authors tested the effectiveness of the same formulation on children's nutritional status using a larger sample size. In study 26, the authors offered daily portions of 30g and 45 for infants 6-12 months and 12-18 months of age, respectively. The intervention was implemented over 12 months.

General remarks

The font should also be uniform (e.g. see text: “The study found that consumption of 25 g cricket powder per day was safe and improved gut health and reduced systemic inflammation”)

Thanks for pointing this out. The font size has been amended accordingly.

Latin words, phrases and abbreviations, including generic and specific names, should be written in italic throughout the text (e.g. Shigella,…).

Thanks for pointing this out. The text has been amended accordingly.

English needs a little improvement - check spelling.

Apologies for the spelling mistakes. Typos were correct and a spelling check was performed.

Round 2

Reviewer 1 Report

line 195, As I know it the phrase you are  looking for is Likert-style rather than  Linkert. There should not be an "n" in that word. 

Author Response

line 195, As I know it the phrase you are  looking for is Likert-style rather than  Linkert. There should not be an "n" in that word. 

Thanks for pointing this typo out. The text has been corrected. 

Reviewer 2 Report

The manuscript is significantly improved. Please, re-check the literature sources.

Author Response

The manuscript is significantly improved. Please, re-check the literature sources.

Thanks for reviewing the manuscript. The literature sources were checked for accuracy.